



# Comparison of in-situ snow depth measurements and impacts on validation of unpiloted aerial system lidar over a mixed-use temperate forest landscape

Holly Proulx[1], Jennifer M. Jacobs[1,2], Elizabeth A. Burakowski[2], Eunsang Cho[1,2], Adam G. Hunsaker[1,2], Franklin B. Sullivan[2], Michael Palace[2,3], Cameron Wagner[1]

[1]Department of Civil and Environmental Engineering, University of New Hampshire, Durham, NH, 03824, USA
[2]Earth Systems Research Center, Institute for the Study of Earth, Oceans, and Space, University of New Hampshire, Durham, NH, 03824, USA
[3]Department of Earth Sciences, University of New Hampshire, Durham, NH, 03824, USA

*Correspondence to*: Jennifer M. Jacobs (Jennifer.jacobs@unh.edu)

**Abstract.** The accuracy and consistency of snow depth measurements depend on the measuring device and the conditions of the site and snowpack in which it is being used. This study compares collocated snow depth measurements from a magnaprobe automatic snow depth probe and a Federal snow tube, then uses these measurements to validate snow depth maps from an unpiloted aerial system (UAS) with an integrated Light Detection and Ranging (lidar) sensor. We conducted three snow depth sampling campaigns from December 2020 to February 2021 that included 39 open field, coniferous, mixed, and deciduous forest sampling sites in Durham, New Hampshire, United States. Average snow depths were between 9 and 15 cm. For all sampling campaigns and land cover types, the magnaprobe snow depth measurements were consistently deeper than the snow tube. There was a 12% average difference between the magnaprobe (14.9 cm) and snow tube (13.2 cm) average snow depths with a greater difference in the forest than the field. The lidar estimates of snow depth were 3.6 cm and 1.9 cm shallower on average than the magnaprobe and snow tube, respectively. While the magnaprobe had a better correlation with the UAS lidar, the root mean square errors were higher for the magnaprobe than the snow tube, likely due to overprobing by the magnaprobe into leaf litter. Even though the differences between the in-situ sampling methods resulted in modest performance differences when used to validate the UAS lidar snow depths in this study, measuring vegetation height, leaf litter, and soil frost with in-situ snow depths from multiple sampling techniques helped to account for the errors of in-situ snow depth for robust validation of the UAS snow depth maps.

**Short Summary.** This study compares snow depth measurements from two manual instruments and an airborne platform in a field and forest. The manual instruments' snow depths differed by 1 to 3 cm. The airborne measurements, which do not penetrate the leaf litter, were consistently shallower than either manual instrument. When combining airborne snow depth maps with manual density measurements, corrections may be required to create unbiased maps of snow properties.

## 1 Introduction

Snow depth is the most commonly measured snow macrophysical property followed by snow presence, snow water equivalent (SWE) and snow bulk density (Pirazzini et al. 2018). While snowpack conditions are important



to both research and operations, it is still challenging to obtain measurements. Snow depth is the easiest snowpack
property to measure in the field and is considered to be an observation that can be measured relatively precisely
without considerable expertise or expense. Hundreds of snow depth measurements can readily be taken in a single
day and automated samplers can considerably increase that number (Sturm and Holmgren 2018). Sturm et al.
(2010) estimated that 20 to 30 snow depth measurements can be made in the time it takes to obtain a single SWE
measurement. Because snow depth is assumed to have greater spatial variability than snow density (Elder et al.
1998), numerous snow depth measurements are often made per snow density measurement then combined to
obtain SWE (López-Moreno et al. 2013). A snow survey usually includes both gravimetric SWE sampling and
snow depth measurements collected over a large area; a technique is referred to as "double sampling" (Derry et
al. 2009; Rovansek et al. 1993). Additionally, estimating SWE from snow depth is considerably easier than
measuring SWE using snow density from snow tubes measurements. Jonas et al. (2009) and Sturm et al. (2010)
developed simple methods to predict the bulk density and, in turn, SWE, based on snow depth measurements, the
day of the year, and snow class, thus entirely eliminating the need to make bulk density measurements.
Subsequently, others have tested and advanced approaches to predict the bulk density and estimate SWE
(Guyennon et al. 2019; Hill et al. 2019).

As reviewed by Kinar and Pomeroy (2015) and Kopp et al. (2019), there are various methods to observe snow
depth including (1) traditional in-situ observations, (2) non-destructive radar, lidar, and Structure from Motion
(SfM) methods, and (3) satellite remote sensing. The latter two methods, which improve the spatial coverage,
typically still rely on in-situ snow depth measurements for calibration of operational technique and validating
remotely sensed observations and model output. Traditional in-situ observations can be measured manually or
automatically. While automated measurements using ultrasonic, laser depth sensors, or time-lapse cameras in
combination with measuring rods are increasing in popularity (Kinar and Pomeroy 2015) (Kopp et al. 2019), in-
situ measurements remain a mainstay of research and operations (Kinar and Pomeroy 2015; Pirazzini et al. 2018).

Manual in-situ snow depth measurements are typically made using snow stakes, rulers, or narrow diameter snow
probes (Kinar and Pomeroy 2015; Pirazzini et al. 2018). Snow tube samplers, which have been in use since the
1930s, also measure snow depth because SWE is the product of snow depth and the depth averaged snowpack
density. The magnaprobe, an automatic snow depth probe that records snow depth and GPS measurements, has
considerably increased the number of georeferenced snow depth observations that can be made in a single day
and is used extensively for snow depth research campaigns (Sturm and Holmgren 2018; Walker et al. 2020). For
these snow depth measurements, the probe is manually pushed through the snow until it hits ground, while the
magnetostrictive basket floats on the snow surface; at the push of a button, the magnaprobe automatically records
the distance between the probe tip and basket. Measurement variability and errors are sometimes reduced by
repeating the measurement, typically three times (Leppänen et al. 2016).

SWE measurement errors associated with snow tube samplers are relatively well understood. Known issues
include biases as compared to snow pit measurements (Dixon and Boon, 2012; Farnes et al., 1983; Goodison,
1978; Sturm et al., 2010), accuracies around +/- 5% to 10% for an individual instrument, and differences among
SWE from different snow tube models (e.g., the Meteorological Service of Canada, the Federal or Mt. Rose, the



Adirondack, and the Snow-Hydro) that can exceed 10% (Farnes et al. 1983). These errors are attributed to issues
in obtaining the correct snow weight due to over- or under-sampling of snow in the core tube and accuracies in
spring or digital balances used to weigh the core.

As compared to snow tube samplers, much less is understood about the errors in snow depth measurements using
snow probes and differences among commonly used measurement techniques. The magnaprobe, which measures
snow depth with a precision of less than 0.1 mm, has the potential for low biases if its basket settles into soft
surface snow (cratering), but those biases are typically less than 1 cm (Sturm and Holmgren 2018). High biases
occur if a snow probe is inserted off vertically or the rod penetrates the substrate (overprobing) (Sturm and
Holmgren 2018). For the former case, reasonable operation will typically insert a rod within 5° of vertical and
result in an error of less than 0.4%, or 0.2 cm for 50 cm deep snow (Sturm and Holmgren 2018). For overprobing,
the error depends on the ground surface and the operation. Solid or frozen ground surfaces have negligible
overprobing. However, unfrozen natural surfaces may have considerable penetration (Derry et al. 2009) with
typical biases on the order of 5 to 10 cm (Berezovskaya and Kane 2007; Sturm and Holmgren 2018). Berezovskaya
and Kane (2007) estimate that snow depth errors cause SWE overestimates of 4 to 20% in northern Alaska.

Emerging remote-sensing methods, terrestrial laser scanning (TLS) (Currier et al. 2019; Grünewald and Lehning
2015), Unpiloted Aerial System (UAS) SfM (Bühler et al. 2016; Harder et al. 2016; Nolan et al. 2015), and UAS
lidar (Harder et al. 2020; Jacobs et al. 2021), can measure snow depth to within a centimeter at high spatial
resolutions. However, validation of those observations is challenging. For example, snow depth observations from
TLS and UAS lidar measurements are biased lower than those from in-situ snow probe observations in the forest
(Currier et al. 2019; Harder et al. 2020; Hopkinson et al. 2004; Jacobs et al. 2021). The causes of these differences
have been partially attributed to the snow probe's ability to penetrate the soil and vegetation and to human
observers who tend to make snow depth measurements in locations with relatively high snow (Sturm and
Holmgren 2018). Results from the comparison between snow depths measured using UAS lidar and a magnaprobe
(Jacobs et al. 2021) implied that the magnaprobe biases were greater than those taken using the Standard Federal
snow tube. Their work suggests that using the Federal snow tube snow depth measurements to validate UAS snow
depth products might be preferable to using magnaprobe measurements.

The goal of this brief study is to determine 1) if the magnitude of the snow depth measurements using a
magnaprobe and a Federal tube are significantly different in an ephemeral snow environment, 2) if the differences
vary by land cover type, 3) the magnitude of forest leaf litter impacts relative to any snow depth differences, and
4) how the two measuring techniques impact UAS lidar snow depth validation. Towards that end, we conducted
three snow depth sampling campaigns from December 2020 to March 2021 over field and forest plots at
Thompson Farm in Durham, New Hampshire, USA. The discussion below describes the results of these
exerperiments.



## 2 Site, Data, and Methods

### 2.1 Study Site

This study was conducted at the University of New Hampshire's Thompson Farm Research Observatory in southeast New Hampshire, United States (N 43.11°, W 70.95°, 35 m above sea level, ASL). The 0.83 km² site has mixed hardwood forest and open field land covers (Burakowski et al. 2018; Burakowski et al. 2015; Perron et al. 2004) that are characteristic of the region (**Fig. 1**). The agricultural fields are managed pasture grass with unmown grass in local areas. The deciduous, mixed, and coniferous forest is composed primarily of white pine (*Pinus strobus*), northern red oak (*Quercus rubra*), red maple (*Acer rubrum*), shagbark hickory (*Carya ovata*), and white oak (*Quercus alba*) (Perron et al. 2004). The forest soils are classified as Hollis/Charlton very stony-fine sandy loam and well-drained; field soils are characterized as Scantic silt-loam and poorly drained (Cho et al. 2021; Perron et al. 2004).

In-situ sampling was conducted at 39 sites located along three parallel transects (**Fig. 1**). The approximately 145 m long transects were laid out from east to west. The transects were separated by approximately 10 m, north to south. From east to west, each transect started in the open field area, then transitioned to the coniferous, then mixed, and finally, deciduous forested areas. Each of the three transects had 13 sampling sites, four sites were in the open field area, three in the coniferous forest, three in the mixed forest, and three in the deciduous forest, that were marked with a stake. The stake locations were geolocated using a Trimble© Geo7X GNSS Positioning Unit and Zephyr™ antenna with an estimated horizontal uncertainty of 2.51 cm (standard deviation 0.95 cm) and 4.17 cm (standard deviation 4.60 cm) for the field and forest respectively after differential correction. Three Cold Regions Research and Engineering Laboratory-Gandahl (CRREL-Gandahl) soil frost tubes (Gandahl 1957; Rickard and Brown 1972; Sharratt and McCool 2005) were located in the field and forest approximately 25 m south of the field transect. UAS lidar surveys were conducted over approximately an 0.2 km² area that encompassed the transects.

### 2.2 *In-Situ* Sampling Methods

Snow depth was measured using a magnaprobe and a Federal snow sampler, also known as a snow tube. The Federal snow tube with its long operational history (Clyde 1932) served as a historical reference against the magnaprobe. A magnaprobe consists of an avalanche probe-like rod of about 1.5 m in length that contains a magnetostrictive device and a sliding magnetic disk-shaped basket with a 25 cm diameter. The rod has a 1.27 cm diameter with an affixed tip that tapers to a point to help penetrate ice layers. The magnaprobe was operated by inserting the pole into a snowpack until the tip of the pole reached the ground surface, allowing the basket to slide down to float on top of the snow. A handheld portable keypad connected to a datalogger recorded the snow depth between the tip of the pole and the bottom of the basket.

A Federal snow sampler is an aluminum tube, about 76 cm in length with a 4.13 cm inner diameter, that is used to measure snow depth and SWE (Clyde 1932). To measure snow depth, the snow tube was inserted vertically into the snowpack until it reached the ground, and a depth was read at eye level. Snow depth was recorded to the nearest 0.5 cm. To measure SWE, the snow tube was then lifted out of the snowpack, using a spatula as needed to ensure that snow did not fall out of the tube. The snow and snow tube were weighed using a digital hanging





scale (CCi HS-6 Electronic Scale, 2 gram resolution). Snow mass was the total mass net of the empty tube mass.
Snow density was determined from the snow mass and sampled volume.

Sampling campaigns were conducted on 18 December 2020, 4 February 2021, and 24 February 2021. A total of
351 paired magnaprobe and Federal snow tube snow depth observations were collected during each campaign. At
each of the 39 sampling locations, nine measurements were made in a 1m x1 m area. Previous UAS lidar snow
depth precision analyses indicated that snow depth differences of 1 cm or less could be detected in a 1x1 m area
using nine samples for most of the study area (Jacobs et al. 2021). At each location, a 1x1 m square polyvinyl
chloride (PVC) grid was placed on the snow surface with one vertex located coincident with a stake. The
orientation of two adjacent sides of the grid was recorded. Nine magnaprobe depth measurements were made at
an approximately even spacing within the 1x1 m grid. Immediately after the magnaprobe measurements, snow
tube snow depth measurements were made at the same nine locations by positioning the snow tube directly over
each magnaprobe sampling location. At a 10[th] location within each 1x1 m grid, the snow tube was used to make
a SWE measurement. For the 24 February 2021 campaign, after the magnaprobe measurements were completed
for the two northern transects, the instrument was transferred to a new operator who made measurements on the
southernmost transect (Transect 1). The QA/QC process identified notable errors for observations from that
transect. Transect 1 data for that date were removed from the analysis.

Moultrie Wingscapes Birdcam Pro Field Cameras were used to capture images of the snowpack relative to a 1.5
meter marked PVC pole following the method used in NASA's 2020 SnowEx field camera campaign in Grand
Mesa, CO (personal communication, 16 November 2020). Three cameras were used; one was in the open field,
one was in the coniferous forest, and one was in the deciduous forest (**Fig. 1**). Each camera was mounted
approximately 0.85 m above the ground and placed approximately 5.5 m from its respective PVC pole. Each
camera's field of view included the entirety of the PVC pole, some of the ground surface below the pole, and
some open area above the pole. Each PVC pole was spray-painted red and was marked with 1 cm and 10 cm
increments. The cameras captured images of the poles every 15-minutes for the duration of the study period. Snow
depth was derived by manual inspection of the photos and recorded to the nearest cm.
**2.3 Ancillary Soils and Vegetation Cover Data**
2.3.1. Soil Frost
Daily soil frost depth data were collected at field and forest locations at the Thompson Farm Research Observatory
using (CRREL-Gandahl) style frost tubes (Gandahl 1957). The frost tubes have flexible, polyethylene inner tubing
filled with methylene blue dye whose color change is easy to differentiate when extruded from ice (Gandahl 1957;
Rickard and Brown 1972; Sharratt and McCool 2005). The outer tubing consists of PVC pipe installed between
0.4 to 0.5 m below the soil surface (Ricard et al. 1976; Sharratt and McCool 2005). The field and forest sites each
had three soil frost tubes. The average soil frost depth at the field and forest sites was calculated for each sampling
day.



2.3.2. Leaf Litter
Leaf litter depth was measured on 2 April 2021 after the spring snowmelt. The leaf litter depth was measured at
each snow depth sample location. Sampling was conducted using a PVC collar or round ring that is 8 cm in depth
and 10 cm in diameter (Kaspari and Yanoviak 2008). The collar was placed in the leaf litter and was pushed down
until it was through the leaf litter layer. If sticks or larger stones were in the way, they were either carefully
removed or the collar was moved slightly to an adjacent location.  Measurements were taken using a wooden ruler
at four cardinal points in the collar.  The four measurements were recorded and averaged, and the final litter depth
value was recorded to the nearest cm.

Magnaprobe penetration depth measurements were also made when snow was not present to capture the probe's
penetration into the leaf litter. Directly following the 2 April 2021 leaf litter sampling using the collar, 20
magnaprobe leaf litter depth measurements were made at each of the 39 snow depth sampling locations.
Measurements were taken within a 1.5 m radius of the stake.  When using the magnaprobe, the weight of the probe
was the only force applied on the ground to minimize penetration into the duff layer and underlying soil. The
probe was gently rested on the ground rather than being forced into the ground.  The 20 measurements were
recorded and averaged to obtain a magnaprobe litter depth at each location.
**2.4 Lidar Sampling**
UAS snow-on lidar surveys were conducted at Thompson Farm prior to in-situ sampling on each of the campaign
dates. A snow-off baseline survey was conducted on 2 April 2 2021 following snowmelt. The sensor payload
consisted of the Velodyne VLP-16 laser scanner, and the Applanix APX-15 Inertial Navigation System (INS;
GPS+IMU). The VLP-16 is a lightweight (~830 grams) low power (~8W) sensor, which makes it ideal for UAS
deployment. The sensor incorporates 16 rotating IR lasers that are arranged and oriented on the payload to provide
a 30° along-track field of view with a cross-track field of view limited only by the range of the sensor
(approximately 100 m). At an altitude of 65 m, the range of the sensor produces an effective cross-track field of
view of approximately 98°, but varies depending on the characteristics of the target surface. Each laser operates
at a wavelength of 903 nm. This wavelength is ideal because it is outside of the first major electromagnetic
absorption feature of snow (centered at 1030 nm). A reduction in signal strength would be observed over snow
cover for lidar sensors that operate at wavelengths coinciding with strong electromagnetic absorption. The VLP-
16 has two return modes, single-return and dual-return, which record the strongest return or the strongest and the
last return, respectively. In dual-return mode, the VLP-16 collects ~300,000 distance measurements per second
with a reported uncertainty of 3 cm at a range of 100 m.

For these acquisition missions, the VLP-16 was hard-mounted to a DJI Matrice 600 to maintain constant lever
arm offsets between the inertial navigation system (INS) GPS antenna, the lidar sensor, and the INS board. As
opposed to a gimbal mounted system, this hard-mounted configuration achieves a more tightly coupled system,
resulting in improved point cloud geolocation accuracy. The lidar sensor was set to dual-return mode to improve
ground detection in the forested areas of our field site. We flew the system at an altitude of 65 m with a flight
speed of 3 m/s and ~40 m spacing between flight lines. Flights produced between a total of ~70-140 million returns
per mission, depending on site ground conditions.




Lidar observations were georeferenced using position and attitude measurements acquired with the Applanix
APX-15 Inertial Navigation System (INS). The INS produced 2–5 cm positional, 0.025 degree roll and pitch, and
0.08 degree true heading uncertainties following post-processing. Post-processing of INS data was performed
using POSPac UAV (v. 8.2.1, Applanix Corporation 2018), correcting differentially against a permanent
Continuously Operating Reference Station (CORS) at the University of New Hampshire in Durham, NH (NHUN).
Position and attitude data were output as a Smoothed Best Estimate of Trajectory (SBET), then time synchronized
with lidar returns to produce a georeferenced point cloud using LidarTools (v. 3.1.4, Headwall Photonics, Inc.).

Three-dimensional point clouds were processed using the progressive morphological filter algorithm (PMF) to
identify ground returns. For ground classification, point clouds were chunked into 100 m square tiles with a 15 m
buffer on all sides using catalog options in lidR to ensure returns near tile edges were classified. PMF was
parameterized using a set of window sizes of 1, 3, 5, and 9 m, and elevation thresholds of 0.2, 1.5, 3, and 7 m,
which were determined by varying value sets and assessing digital terrain models (DTMs) to determine the
parameter sets that produced a visually smooth surface over a dense grid (Muir et al. 2017). Following ground
classification for each tile, returns within the 15 m tile buffers were removed, and all resulting 100 m square
ground classified tiles were merged. The result of the PMF is that non-ground returns (i.e., trees, shrubs, and
noise) were filtered out of the point cloud data sets, so that only returns from ground surfaces remained. The two
data sets, non-ground returns and ground returns from the original point clouds, were coded according to LAS
specifications and merged. The ground returns were extracted for the 1 x 1 m square sampling sites, corresponding
to the alignment and orientation of the respective PVC grids. The lidar snow depth was calculated as the difference
between the mean snow-on and mean snow-off elevations within each sampling grid.
**2.5 Statistical Approach**
The magnaprobe, snow tube, and lidar snow depth measurements were summarized and compared for the field
and forested areas by sampling campaign date following (Willmott 1982). Each comparison was conducted using
the individual grid cell measurements (N = 9 at each grid cell), and grid cell average depths. Sample statistics that
were calculated and compared for each of these datasets included the mean and standard deviation, the bias, the
mean absolute error (MAE), and the root mean square error (RMSE). A line of best fit was generated for each to
provide the corresponding slopes and intercepts, and r-squared values. As described by Willmott (1982), MAE of
the compared data sets, is given as:
$$MAE = N^{-1} \sum_{i=1}^{N} |X_j - Y_j| \qquad (1)$$

where X and Y are two of the magnaprobe, snow tube, and lidar snow depth and N is the number of samples. The
root mean square error (RMSE) is the average squared difference between the compared data sets given as:
$$RMSE = [(N^{-1}) \sum_{i=1}^{N} (X_i - Y_i)^2]^{0.5} \qquad (2)$$

The mean difference between snow depth from two sampling techniques quantifies the bias between the
measurements and, in doing so, identifies whether one sampling technique yields deeper or shallower snow on
average than another technique. The standard deviation characterizes the variability of those individual



differences. Ideally, measurements from the two instruments would have little to no systematic bias and the snow
depth differences would be relatively consistent at each sampling location. The RMSE of the snow depth
differences combines bias and variability into a single metric. Finally, significance tests of the mean snow depth
differences were conducted for each grid cell using t-tests after testing the normality of the data. The 24 February
2021 campaign, in which all measurements from Transect 1 were omitted from the dataset due to sampling errors,
had a lower sample size than other sampling campaign dates.

**3 Results and Discussion**

**Table 1** summarizes the snow and soil conditions by sampling campaign. Between the December and the 4
February sampling campaigns, there was a melt event during mid-December in which the entire snowpack ablated.
The next significant snowfall event (15 cm) occurred on 1 February 2021. The snowpack experienced little
additional accumulation or ablation between 4 February and 24 February. The field camera observations indicate
that the snowpacks had similar depths, between 10 and 15 cm, on the three sampling dates with modestly deeper
snow in the field than the forest. The February snowpack density values ($0.15 - 0.24$ g/cm$^3$) were higher than
those in December ($\sim 0.10$ g/cm$^3$). There was limited soil frost (< 4 cm) during the early winter December
campaign in the forest and the field. The deepest soil frost was on 4 February 2021 with 15.1 cm in the field and
5.9 cm in the forest, with similar soil frost conditions on 24 February 2021.

**3.1 Magnaprobe vs. Snow Tube**

The full experiment yielded individual 936 pairs of snow depth measurements from the snow tube and the
magnaprobe (**Fig. 2a**). Overall, there was moderate agreement ($R^2 = 0.55$) between the two datasets for all three
sampling campaigns (**Table 2**). The snow depths measured by the magnaprobe (14.9 cm average snow depth)
were deeper than the snow tube (13.2 cm average snow depth) with an overall bias of 1.7 cm. The magnaprobe
snow depth was at least 0.5 cm deeper than the snow tube in 74% of the 936 measurement pairs. Only 6.3% of
the pairs had snow tube snow depths exceeding magnaprobe snow depths by 0.5 cm or more. 7.4% of the pairs'
magnaprobe snow depths were over 5.0 cm deeper than the snow tube. In eight pairs of measurements, the
magnaprobe snow depth was more than double the snow tube snow depth.

Out of the nine paired sampling locations in each grid, the majority (an average of 8.7, 7.7, and 7.0 of the sampling
locations in each grid on 18 December 2020, 4 and 24 February 2021, respectively) had magnaprobe snow depth
values that were deeper than those measured using the snow tube. The magnaprobe snow depth values were
significantly greater than those measured using the snow tube for 39 and 31 of the 39 sampling locations on 18
December 2020 and 4 February 2021, respectively, but only 11 out of the 26 sampling locations on 24 February
2021. The mean differences were 2.3, 1.4, and 1.6 cm, with RMSE values of 3.0, 2.3, and 3.3 cm, on 18 December
2020, 4 and 24 February 2021, respectively, which is on the order of 15 to 25% of the overall depth observed
during these campaigns. Despite the biases, the average within cell snow depth variability was nearly identical
for the magnaprobe and the snow tube in the field (1.3 cm standard deviation for the magnaprobe). In the forest,
the magaprobe's 2.0 cm within cell standard deviation modestly exceeded the snow tube's 1.5 cm standard





deviation. The slightly reduced agreement on 2/24 may be due to a 1-4 cm thick ice layer at the bottom of the
snowpack in local depressions.

The overall agreement between the snow tube and magnaprobe was better when the nine measurements within a
single 1x1 m grid cell were averaged at each of the sampling locations (**Fig. 2b** and **Table 2**). There is a notable
improvement in grid cell statistics, and the correlation is stronger (overall $R^2 = 0.76$), with slopes closer to one,
intercepts closer to zero, and the RMSE values reduced to 2.5 cm or less. Although averaging has no impact on
the overall bias, the range of differences among pairs narrowed. Boxplots show that there is a consistent difference
(magnaprobe minus snow tube) that is typically constrained to less than 3 cm, but that a limited number of outliers
were observed (**Fig. 3b**). The magnaprobe snow depth was at least 0.5 cm deeper than the snow tube in almost all
grid cells (86.7%), but only three grid cells had differences greater than 5 cm. There were no instances in which
there was a doubling of snow depth.
**3.2 Magnaprobe vs. Snow Tube by Land type**
The magnaprobe and snow tube snow depths differ by land type, with the field having deeper snow and more
spatial variability than the forest land types (**Fig. 4**). Among the three forest types, the deepest snow was in the
deciduous-dominated forest, with mixed and coniferous forest having similar snow depths. The mean difference
between the magnaprobe and snow tube snow depths is a modest 1.3 cm in the field and a 1.9 cm in the forest,
with differences of 1.9, 2.0, and 1.9 cm in the deciduous, mixed, and coniferous land types, respectively. Based
on t-test results, the magnaprobe measured significantly deeper snow depth compared to the snow tube in both the
field and the forest. The t-test results identified significant differences between snow depths from the two probing
techniques regardless of whether individual locations (p-value < 0.001) or grid cell average snow depths (p-value
= 0.02) were used. Based on Welch's adjusted ANOVA test, there are no significant differences in overprobing
among forest land types (p-value = 0.24). The RMSE values between the magnaprobe and snow tube snow depths
are 3.0 cm (2.3 cm) and 2.5 cm (2.0 cm) for the forest and field sampling sites (grid average values), respectively.
Thus, the sampling method has a different impact in the field than the forest and the RMSE and bias values provide
an indicator of the different errors associated with in-situ measurements based on land type when used for model
or remote sensing validation.
**3.3 Impacts of Leaf Litter on Magnaprobe vs. Snow Tube Depth**
The range of leaf litter depths measured in the forest using the collar was typically 3 to 7 cm with an average leaf
litter depth of 3.9 cm (**Fig. 5**). The snow-off magnaprobe litter depth measurements in the forest had an average
value of 5.8 cm and the differences were significantly larger than depths measured using the collar (p-value <
0.001). The litter depths in the forest regardless of measurement technique exceeded the differences between the
magnaprobe and snow tube snow depths in the forest, which were 2.5, 1.7, and 1.4 cm on 18 December, 4
February, and 24 February, respectively.
**3.4 Lidar and *In-Situ* Snow Depth Comparison**
While the previous sections identified significant differences between the magnaprobe and snow tube snow
depth measurements, the average differences, 1.3 and 1.9 cm in the field and forest, respectively, are



relatively modest. One of the motivations for this study was to understand the impact of those differences
on the validation of emerging high resolution snow depth datasets such as those from UAS SfM or lidar
observations. Here, we briefly examine the lidar snow depth performance relative to both in-situ sampling
techniques and land type (**Table 3** and **Fig. 6**), then discuss the impact of different sampling techniques on
that evaluation.
The lidar-derived snow depths for each of the 1x1 m grid cells were extracted as described in Section 2.2.
For both magnaprobe and snow tube measurements, the agreement with lidar is markedly better in the field
than the forest (**Fig. 6**). Overall, the lidar estimates of snow depth are typically shallower than the in-situ
observations (**Table 3**). This is particularly evident for the 24 February 2021 forest lidar snow depths. The
lidar also has larger cell-to-cell variability than the in-situ measurements, as quantified by the standard
deviation, particularly in the forest. This large variability in the forest combined with the relatively small
range of snow depths even across sampling dates makes it nearly impossible to identify relatively shallow
or deep snow depths within the forest. The very low correlation values for both in-situ validation approaches
reflect the low signal-to-noise ratio. In contrast, there is fairly strong evidence in the field that snow depth
differences that exceed 3 cm are discernible.
**Fig. 7** shows that the differences between the lidar and in-situ observations, regardless of method, are
considerably larger than the differences between the two in-situ sampling methods. The magnaprobe's
potential to overprobe through leaf litter and duff layers to a greater extent than the snow tube impacts the
quantification of performance. Overprobing negatively impacts the bias, MAE, RMSE, and linear regression
intercept metrics. The RMSE values are slightly higher for the magnaprobe than the snow tube, and to a
large extent this reflects the higher bias when using the magnaprobe as compared to the snow tube. In
contrast, the snow tube's RMSE is largely due to the snow tube's high site to site differences rather than an
overall bias. Thus, for individual locations, the magnaprobe is more consistent in its agreement with the
lidar. This is also reflected in the higher $R^2$ value.

## 4 Discussion

### 4.1 Uncertainty and impacts from overprobing

This study quantifies the differences between snow depth measurements made with a magnaprobe and with a
Federal snow tube sampler. The differences seem to be primarily associated with greater overprobing by the
magnaprobe into vegetation/organic layers and thawed soils. The result was that magnaprobe snow depth
measurements were observed to be higher than snow tube measurements, with a greater difference in the forest
than the field. This result agrees with previous studies. An average of 5 cm high bias occurred in the tundra matte
during the Cold Land Processes Experiment (CLPX) Alaska campaign (Sturm and Holmgren 2018). A 2018
experiment in a single snow pit within an open tundra environment found a 7.6 cm average overprobe penetration
(Canada, 2018). Using a snow probe, Berezovskaya and Kane (2007) found a 5 to 9 cm bias in northern Alaska.



They also noted that overprobing was greater with the probe as compared to the snow tube. The current study's
snow-off magnaprobe forest litter depth measurements of 5.8 cm are similar to these previous finding.

Sturm and Holmgren (2018) suggested that operators need to learn to push a magnaprobe through snow yet not
impale it too deeply into underlying vegetation/organic layers by developing a sense for the base of the snowpack.
However, this recommendation could be difficult to implement over soft vegetation (e.g. tundra) where the probe
easily penetrates the vegetation. In that case, a consistent way to push a magnaprobe is needed by operators,
though any two operators will likely apply a different force (Berezovskaya and Kane 2007). If operators overprobe
it into the base of the (frozen) soils, one should consistently measure the depths in the same way (which would be
snow depth *plus* vegetation) and then subtract typical vegetation depths in the study area from the depths.
Measurements of leaf litter or vegetation depths may help to account for the overprobing errors of magnaprobe
snow depth measurements.

Overprobing also impacts SWE estimates. Given the efficiency of making snow depth measurements, a snow
survey will often make numerous snow depth measurements per snow density measurement then combine the
measurements to obtain SWE (Elder et al. 1998; López-Moreno et al. 2013). In some cases, only snow depth is
measured and bulk density is derived from empirical relationships. In either case, any biases in snow depth will
be transferred to the SWE estimates. Based on leaf litter measurements and the differences between the lidar snow
depth estimates and the in-situ measurements, it appears that both instruments overprobe to some extent. In fact,
a typical application of the snow tube will overprobe by design to extract the snow core and a "plug of soil".
However, because the operator removes any vegetation and soil prior to recording measurements, snow tube
measurements can readily correct for the overprobing. The errors incurred by combining magnaprobe
measurements with snow tube density values to determine SWE likely equal or exceed those from the 1.9 cm
depth differences observed in this study.
**4.2 Recommendations for sampling strategy to validate UAS-based data**
To validate high-resolution snow depth measurements from UAS-based lidar and SfM photogrammetry, reliable
ground-based observations with an appropriate sampling strategy are required. From the surveys conducted in this
study, there are several technical lessons for researchers who will conduct UAS snow depth surveys.

UAS-based snow depth measurements are typically gridded outputs (1-m grid in this study). As compared to using
single measurements along a transect for validation, using the average of multiple-point samples within a grid can
reduce the point-to-point variability and spatial representativeness errors. To test if using fewer in-situ sampling
points makes a difference in the reported performance of the UAS Lidar measurements, the summary statistics
(**Table 3**) were recalculated by randomly sampling one point and three points per grid cell, respectively, and
extracting the paired magnaprobe and snow tube depths. In all cases, the correlations with lidar snow depth
degraded modestly. For example, the nine snow tube samples $R^2$ value of 0.40 decreased to 0.39 and 0.37 for
three and one sample, respectively, and the nine magnaprobe samples $R^2$ value of 0.53 decreased to 0.50 for both
three and one sample, respectively. RMSE values typically increased by 0.3 cm or less with decreasing sample





size. Another challenge with transect style measurements is that it is difficult to capture their locations at the
resolution needed to align the UAS measurements.

It may be advisable to use multiple sampling techniques, rather than a single method, in order to cross-check on
ground snow depth measurements because the measurement errors vary by sampling methods and surface
conditions (e.g. low vegetation, leaf litter, and soils), particularly in shallow snowpacks. As observed in this study,
leaf litter and soil frost can differentially impact in-situ snow depth sampling methods. The 3.9 cm forest leaf litter
depth was nearly double the 2.0 cm snow depth differences. Distinct contributions of forest leaf litter depth to
magnaprobe and snow tube snow depths may occur because the narrow magnaprobe fully penetrates the leaf litter
and the larger diameter snow tube only partially penetrates the litter, or the magnaprobe may only partially
penetrate the leaf litter but the snow tube does not break through the leaf litter. Partial penetration of the
magnaprobe into the leaf litter layer (i.e., overprobing) may vary by the freeze-thaw state of the duff layer and/or
mineral soil layers beneath the leaf litter layer. The horizontally aligned, matted leaf litter could also limit snow
tube penetration. High spatial variability of leaf litter depth could also be a factor, though this was not quantified
here. Thus, differences among in-situ methods in forested areas point to the particular importance of in-situ
validation in forested areas and, more generally, sampling with multiple methods in an area with a nonuniform
underlying substrate.

Emerging techniques such as automated snow depth retrievals from field cameras may offer improved validation
for high resolution remote sensing observations of snow status. For example, the field camera method outlined
above has a potential to measure snow depth consistently over time. Our preliminary results (Hunsaker et al. 2021)
suggest that snow depth measurements from field cameras may have better agreement with lidar-based snow
depths. An added advantage of field cameras is that the snowpack would not be impacted through destructive
measurements and foot tracks to measurement locations.

### 4.3 Future perspectives

While airborne-based lidar and SfM photogrammetry have been widely used to generate spatially distributed snow
depth maps at scales between ground measurements and satellite or regional snow products (Deems et al. 2013;
Painter et al. 2016), the airborne systems have limited availability for repeated deployments over a season due to
costs, limiting its use for many studies. For field-scale hydrological and ecological research where higher spatial
and temporal resolution snow information are needed, UAS-based lidar and SfM platforms can bridge between
ground measurements and the airborne-based information (Cho et al. 2021). Due to the economic feasibility and
availability of deployment, these systems are increasingly being used and have potentials to advance snow science.
For example, the extent and periods of shallow and ephemeral snowpack will likely increase in a warmer climate
making accurate measurements increasingly (Siirila-Woodburn et al. 2021) important. The UAS observations may
also allow small changes in deeper snowpacks to be observed and, in turn, offer improved understanding of the
snowpack accumulation, ablation, and redistribution. However, as remotely sensed observations or model outputs
continue to improve the ability to estimate snow depth, we appear to be reaching current limits to validate those
improvements.



The current results provide further support for previous studies that articulated the limits of in-situ observations.
For snow study, the UAS lidar-based measurements may be more representative of snowpack conditions than the
point sampling observations used to validate other remotely sensed and modelled snow products. At the same
time, we expect that magnaprobes, or similar high efficiency snow depth measuring techniques, will continue to
be needed for validation of remote sensing and model estimates as an essential and consistent approach. Further
studies in other environments with different vegetation and soil conditions would help to minimize the errors from
in-situ sampling and improve the needed validation of the UAS snow depth maps.
**5 Conclusion**
Manual in situ sampling snow depth measurements can be made quickly and easily, but making consistent,
representative, and unbiased measurements can be challenging when the surface is irregular, vegetation/organic
layers and unfrozen soils result in overprobing, and the leaf litter compacts during the winter. This study quantified
the differences between snow depth measurements made with a magnaprobe and a Federal snow tube and assessed
impacts on the validation of UAS lidar-based snow depth measurements in a mixed-use temperate forest landscape
with ephemeral snowpack. For all sampling campaigns and land cover types, the magnaprobe snow depth
measurements (mean 14.9 cm) were consistently deeper than the snow tube measurements (13.2 cm), which was
a 12% average difference with a greater difference in the forest than the field. The lidar-based snow depths were
shallower than the magnaprobe and snow tube measurements by 3.6 cm and 1.9 cm on average, respectively.
RMSE values between the magnaprobe and the lidar snow depths (3.6 and 5.4 cm) were larger than that between
the snow tube and the lidar (3.2 and 4.4 cm for field and forest, respectively) partially due to overprobing by the
magnaprobe into leaf litter and surface soils. For a robust validation of the UAS lidar and SfM-based snow depth
maps, there are several suggestions for those who conduct similar studies.
1)  For validation of the lidar snow depths, the use of the average of multiple-point samples within a grid is
476       recommended instead of single measurements, because the average of multiple-point samples can reduce the
477       point-to-point variability and spatial representativeness errors.
2)  Measurements of leaf litter and soil frost may help to account for the overprobing errors, particularly when
479       using a magnaprobe.
3)  To cross-check on ground snow depth measurements, the use of multiple sampling techniques is highly
481       recommended (rather than a single method) because the measurement errors vary by sampling methods and
482       surface conditions (e.g., low vegetation, leaf litter, and soils), particularly in shallow snowpacks.
As the UAS lidar or optical systems are increasingly used in snow research, it is prudent to recognize that snow
depth maps produced by these remote sensing products are likely to be modestly shallower than coincident in situ
observations. The differences among measurement techniques in this present study reflect the current study area,
surface conditions for a single season, and the operation of the instruments by this project team. Further studies
to minimize the errors from in-situ sampling in various snow environments in with different vegetation and soil
conditions are needed to accurately validate UAS snow depth maps and to provide guidance on best practices for
using these maps in combination with in situ measurements to represent differences in snow depth and SWE over
space and time.



**Acknowledgements**
This material is based upon work supported by the Broad Agency Announcement Program and the Cold Regions
Research and Engineering Laboratory (ERDC-CRREL) under Contract No. W913E518C0005 and
W913E521C0006. The authors are grateful to Lee Friess for providing a technical review of the draft manuscript,
Mahsa Moradi Khaneghahi for supporting manuscript preparation, and Brigid Ferris for training the team on litter
depth sampling.
**Data Availability**
The UAS-based lidar point clouds and in-situ snow observations are available from the corresponding author upon
reasonable request.
**Author Contributions**
HP, JJ, EB, AH, FS, MP, and EC designed the research.  HP, CW, JJ, AH, FS, MP, EB, and EC conducted field
work to obtain lidar and/or in-situ snow observations. HP, CW, JJ, EB, AH, and MP and performed the analysis.
HP, EC, and AH produced the figures. HP, JJ, EB, and EC wrote the initial draft. All authors contributed to
manuscript review and editing.
**Competing Interests**
The authors declare that they have no conflict of interest.

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

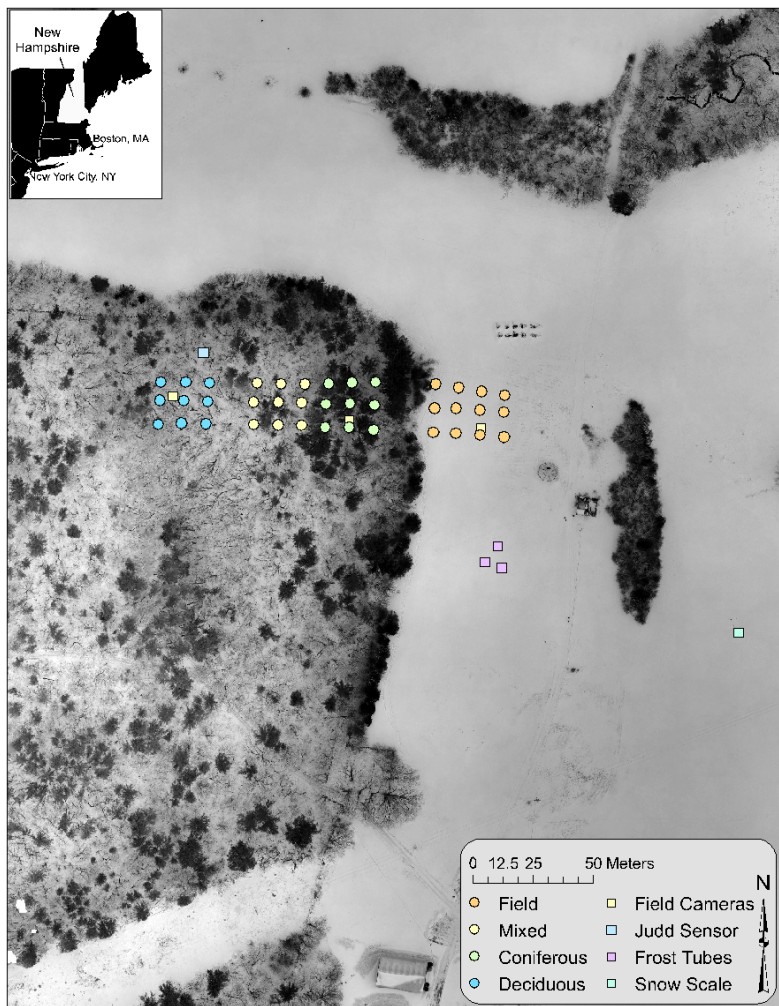


**Figure 1:** The 4 February 2021 aerial optical image of Thompson Farm, Durham NH, USA showing both forest
and field region with snow sampling sites in the field, coniferous, mixed, and deciduous forested areas as well as
the locations of the CRREL-Gandahl soil frost tubes; and field cameras.



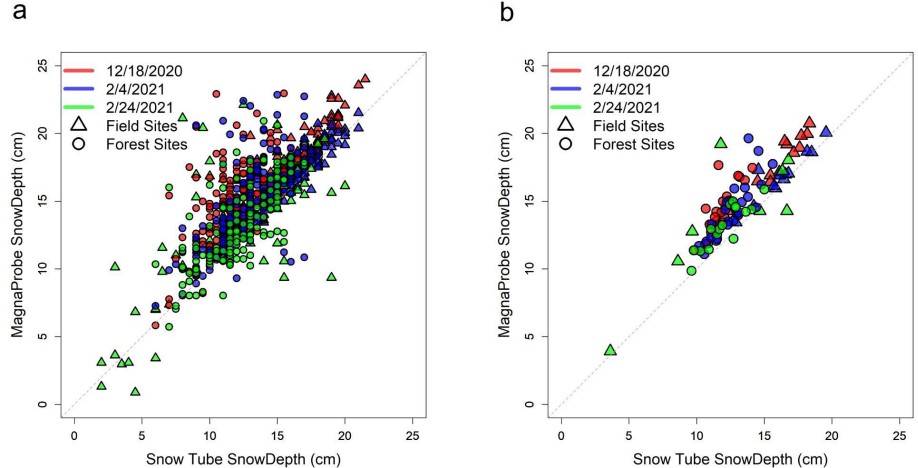

**Figure 2:** Comparison of snow depths measured by magnaprobe and snow tube for the three sampling campaigns using (a) the sampling individual points (n = 936) and (b) using grid cell average values (n=104).

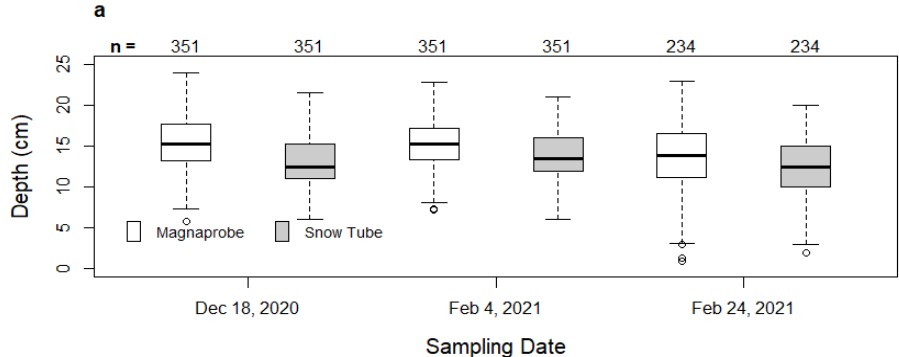

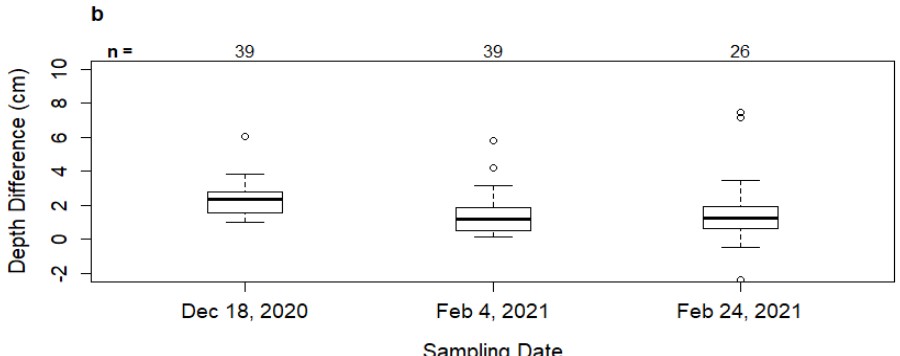

**Figure 3:** Boxplots of snow depths measured by magnaprobe and snow tube for the three sampling campaigns using (a) all the grid cell values and (b) differences between grid cell average values by date where n is the number of (a) sample points and (b) sample grids.



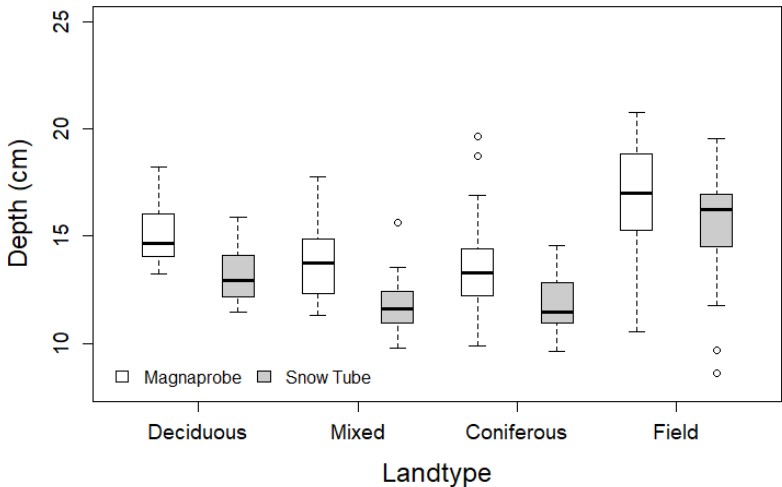

604 .

**Figure 4**: Boxplots of snow depths by land type measured by the magnaprobe and the snow tube for the three
sampling campaigns using the grid cell average values.

607

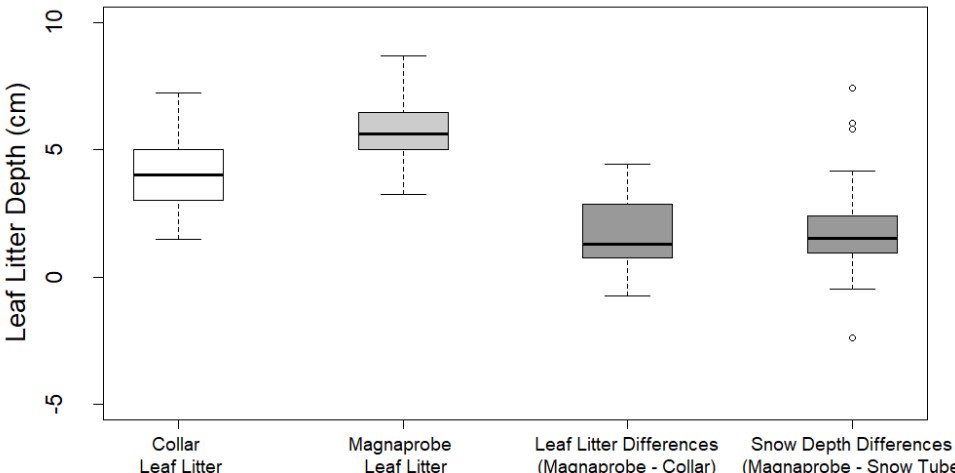

608

**Figure 5**: Boxplots of leaf litter depth measurements taking under snow free conditions on 2 April 2021 by the
leaf litter collar technique and the snow off magnaprobe technique, as compared to boxplots of litter depth
differences as measured by collar and magnaprobe techniques, and snow depth differences measured by
magnaprobe and snow tube for the three sampling campaigns using the grid cell average values in the forest.





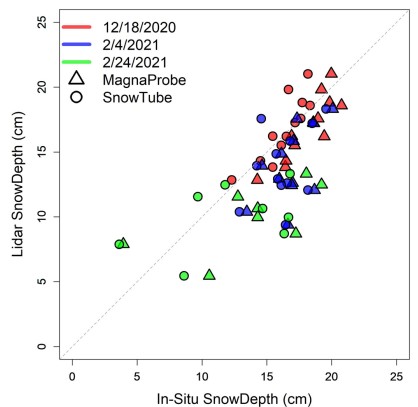
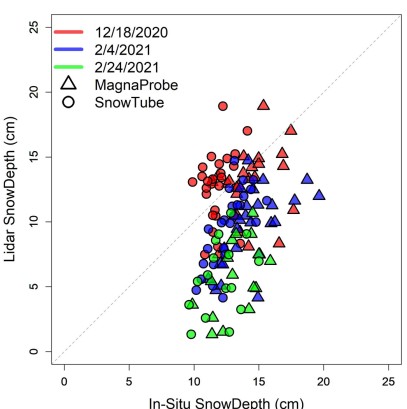


**Figure 6**: Comparison of 1 m grid cell average snow depths measured by the magnaprobe and snow tube versus
the UAS lidar for the three sampling campaigns in the field (left) and the forest (right).

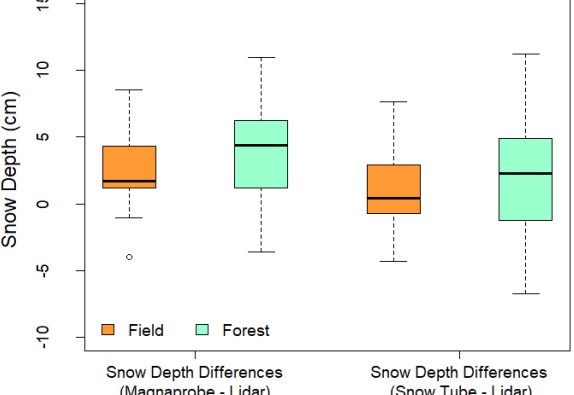


**Figure 7:** Difference of 1 m grid cell average snow depths measured by the magnaprobe and lidar for the three
sampling campaigns in the field and the forest.

**Table 1:** Summary of snow and soil frost conditions during the winter 2020/2021 field campaigns at Thompson
Farm, Durham NH.  Snow depth was measured from field cameras.

| Variables | Land type | Campaign Date | | |
|---|---|---|---|---|
| | | 18 December | 4 February | 24 February |
| Snow Depth (cm) | Field | 10 | 12.5 | 15 |
| | Forest | 9.8 | 10.8 | 9.3 |
| Snow Density (g/cm³) | Field | 0.09 | 0.15 | 0.24 |
| | Forest | 0.10 | 0.15 | 0.20 |
| Soil Frost (cm) | Field | 3.7 | 15.1 | 13.8 |
| | Forest | 2.2 | 5.9 | 2.1 |



**Table 2**: Summarized statistics of snow depths for the magnaprobe and snow tube techniques by the individual
points and the grid cell averaged values for each of the sampling campaign dates. All units are cm except slope
and $R^2$, which are dimensionless.

| Date | Magnaprobe Mean (Std) | Snow tube Mean (Std) | Bias | N | Intercept | Slope | $R^2$ | MAE | RMSE |
|---|---|---|---|---|---|---|---|---|---|
| | | | All Measurements | | | | | | |
| 18 December | 15.5 (3.1) | 13.2 (2.9) | 2.3 | 351 | 1.85 | 0.73 | 0.62 | 2.4 | 3.0 |
| 4 February | 15.2 (2.8) | 13.9 (2.7) | 1.4 | 351 | 2.70 | 0.73 | 0.59 | 1.6 | 2.3 |
| 24 February | 13.6 (3.8) | 12.2 (3.4) | 1.4 | 234 | 4.29 | 0.58 | 0.43 | 2.2 | 3.3 |
| All Dates | 14.9 (3.3) | 13.2 (3.0) | 1.7 | 936 | 3.09 | 0.68 | 0.55 | 2.0 | 2.9 |
| | | | Grid Cell Averages | | | | | | |
| 18 December | 15.5 (2.6) | 13.3 (2.5) | 2.3 | 39 | -0.67 | 0.90 | 0.85 | 2.3 | 2.5 |
| 4 February | 15.2 (2.3) | 13.8 (2.3) | 1.4 | 39 | 0.71 | 0.86 | 0.74 | 1.4 | 1.8 |
| 24 February | 13.6 (3.0) | 12.2 (2.7) | 1.4 | 26 | 2.05 | 0.74 | 0.66 | 1.7 | 2.3 |
| All Dates | 14.9 (2.7) | 13.2 (2.5) | 1.7 | 104 | 0.91 | 0.82 | 0.75 | 1.8 | 2.2 |


**Table 3**: Summary statistics of 1 m grid cell average snow depth values for the lidar as compared to the in-situ
magnaprobe and snow tube separated into forest and field locations. All units are cm except slope and R2, which
are dimensionless.

| Land type | Technique | In-situ Mean (Std) | Lidar Mean (Std) | Bias | N | Intercept | Slope | $R^2$ | MAE | RMSE |
|---|---|---|---|---|---|---|---|---|---|---|
| Field | Magnaprobe | 16.6 (3.3) | 14.1 (3.7) | 2.5 | 32 | 0.38 | 0.82 | 0.53 | 2.9 | 3.6 |
| | Snow Tube | 15.3 (3.2) | | 1.2 | 32 | 2.98 | 0.72 | 0.40 | 2.4 | 3.2 |
| Forest | Magnaprobe | 14.1 (1.9) | 9.9 (3.9) | 4.2 | 66 | -3.28 | 0.94 | 0.21 | 4.5 | 5.4 |
| | Snow Tube | 12.2 (1.3) | | 2.3 | 66 | 0.74 | 0.75 | 0.07 | 3.6 | 4.4 |
