# Peer review of "Comparison of in-situ snow depth measurements and impacts on validation of unpiloted aerial system lidar over a mixed-use temperate forest landscape"

_The Cryosphere, 2022_

## Referee Comment (RC1)

**Review comments** on tc-2022-7 manuscript, entitled," Comparison of in-situ snow depth measurements and impacts on validation of unpiloted aerial system lidar over a mixed-use temperate forest landscape".

**General comments**:

The tc-2022-7 manuscript, entitled," Comparison of in-situ snow depth measurements and impacts on validation of unpiloted aerial system lidar over a mixed-use temperate forest landscape" presents validation of snow depth maps from an unpiloted aerial system (UAS) with an integrated Light 16 Detection and Ranging (lidar) sensor using snow depth 14 measurements from a magnaprobe automatic snow depth probe and a Federal snow tube.

The objectives of the presented work are

- to investigate effects of an ephermeral snow environment, land cover type and forest leaf litter on snow depth measurements using a magnaprobe and a Federal tube are significantly different in an ephemeral snow environment,
- to investigate impacts of validation UAS lidar with mangaprobe and a Federal tube snow measurements.

As general comment, the manuscript is designed and written well.  It consists of all results and conclusions and objectives of the work planned.

The point I miss is how those results on the objectives would differ in case of deeper snowpack. It seems that the snowpack is very shallow like 10-15 cm. What happens if it varies up to 100 cm.  The conclusions should be confined under limited snow depth conditions like up to 15 cm.

Because if snow depth is deeper then they may be some other challenges and errors can occur on the snow measurements by magnaprobe and snow tube. Please have a look at

 López-Moreno, J. I., Leppänen, L., Luks, B., Holko, L., Picard, G., Sanmiguel-Vallelado, A., Alonso-González, E., Finger, D. C., Arslan, A. N., Gillemot, K., Sensoy, A., Sorman, A., Ertaş, M. C., Fassnacht, S. R., Fierz, C., and Marty, C.: Intercomparison of measurements of bulk snow density and water equivalent of snow cover with snow core samplers: Instrumental bias and variability induced by observers, Hydrol. Process., 34, 3120–3133, https://doi.org/10.1002/hyp.13785, 2020.

There is also a recent work published on retrieval of snow depth using webcam images in Cryosphere. It would be good to have look at it as well:

Bongio, M.; Arslan, A.N.; Tanis, C.M.; De Michele, C. Snow depth time series retrieval by time-lapse photography: Finnish and Italian case studies. Cryosphere 2021, 15, 369–387.

---

## Author Comment (AC1)

Referee #1

*Thank you for the original review and the recommendations for minor revisions. We have provided detailed responses to the reviewer in **bold** following each of the reviewer's comments.*

**Review comments** on tc-2022-7 manuscript, entitled," Comparison of in-situ snow depth measurements and impacts on validation of unpiloted aerial system lidar over a mixed-use temperate forest landscape".

**General comments:**

The tc-2022-7 manuscript, entitled," Comparison of in-situ snow depth measurements and impacts on validation of unpiloted aerial system lidar over a mixed-use temperate forest landscape" presents validation of snow depth maps from an unpiloted aerial system (UAS) with an integrated Light 16 Detection and Ranging (lidar) sensor using snow depth 14 measurements from a magnaprobe automatic snow depth probe and a Federal snow tube.

The objectives of the presented work are

- to investigate effects of an ephemeral snow environment, land cover type and forest leaf litter on snow depth measurements using a magnaprobe and a Federal tube are significantly different in an ephemeral snow environment,
- to investigate impacts of validation UAS lidar with mangaprobe and a Federal tube snow measurements.

As general comment, the manuscript is designed and written well. It consists of all results and conclusions and objectives of the work planned.

**Thank you.**

The point I miss is how those results on the objectives would differ in case of deeper snowpack. It seems that the snowpack is very shallow like 10-15 cm. What happens if it varies up to 100 cm. The conclusions should be confined under limited snow depth conditions like up to 15 cm.

**Investigating measurements of a shallow snowpack was the main goal of this research. We agree that it must be specified that the results of this research examined shallow snowpacks. The following sentence was added to the abstract "Shallow snowpacks are often difficult to estimate. Our study was conducted during periods of shallow snowfall to examine the biases both for in-situ field measurements and lidar based estimates." The following sentence will be added to the concluding paragraph in introduction which describes the study goals and hypothesis "While these biases between the two sampling techniques may be negligible in a deeper snowpack, their differences are more significant relative to a shallow snowpack." The shallow snowpack is also mentioned in line 488.**

Because if snow depth is deeper then they may be some other challenges and errors can occur on the snow measurements by magnaprobe and snow tube. Please have a look at

López-Moreno, J. I., Leppänen, L., Luks, B., Holko, L., Picard, G., Sanmiguel-Vallelado, A., Alonso-González, E., Finger, D. C., Arslan, A. N., Gillemot, K., Sensoy, A., Sorman, A., Ertaş, M. C., Fassnacht, S. R., Fierz,

C., and Marty, C.: Intercomparison of measurements of bulk snow density and water equivalent of snow cover with snow core samplers: Instrumental bias and variability induced by observers, Hydrol. Process., 34, 3120–3133, https://doi.org/10.1002/hyp.13785, 2020.

**Many thanks for pointing us to this reference. It is an interesting and unique article that all snow samplers should read. We plan to reference the article and its snow depth findings in the introduction and discussion as follows:**

**Introduction "Lopez-Moreno et al.'s (2020) comparison of nine snow core samplers found that snow depths were relatively consistent when taken over a paved surface. However over uneven ground, the snow depth differences between samplers was much greater and replicate snow depth measurements had a larger variability when compared to the snow density."**

**Discussion "We also agree with Lopez-Moreno et al.'s (2020) finding that it is important to understand the snowpack and conditions for which an individual sampler was designed in order to select the most appropriate sampler."**

There is also a recent work published on retrieval of snow depth using webcam images in Cryosphere. It would be good to have look at it as well:

Bongio, M.; Arslan, A.N.; Tanis, C.M.; De Michele, C. Snow depth time series retrieval by time-lapse photography: Finnish and Italian case studies. Cryosphere 2021, 15, 369–387.

**Thank you for another relevant reference. We plan to add information from the reference to the methods review in the introduction and the discussion section 4.2.**

**Introduction "Automated measurements that include ultrasonic methods, laser depth sensors, and time-lapse cameras utilizing a measuring rods are increasing in use (Kinar and Pomeroy 2015; Kopp et al. 2019 ; Bongio et al. 2021), in-situ measurements remain the mainstay of data collection for research and operations (Kinar and Pomeroy 2015; Pirazzini et al. 2018)."**

**Discussion "Emerging techniques such as automated snow depth retrievals from field cameras may offer improved validation for high resolution remote sensing observations of snow status when the field camera images are clear, and the camera and stake are properly aligned and not prone to movement (Bongio et al. 2021)."**

---

## Author Comment (AC2)

*Thank you for your review and the recommendations for minor revisions. We have provided detailed responses to the reviewer in **bold** following each of the reviewer's comments.*

**General comments:**

Proulx et al. 2021 provide and interesting analysis of the impacts of manual snow depth probing techniques and the implications of that in UAS-lidar snow depth validation. Overall this manuscript is well written and clear in its intentions and execution and was an easy read and I commend the authors on that aspect. Overall I struggle with whether the results have the significance to merit the level of a research article ("substantial and original scientific results") level versus a brief communication (or what I would like to call a technical note but is not an option in TC). The findings do have implications on further work in the area but are not by themselves novel (ie a main conclusion of magnaprobe oversampling versus a snow tube is references as a finding of Berezovskaya and Kane (2007) while the authors and others have already published on uas-lidar snow depth validation and accuracy assessment at this site). In terms of content I have a number of minor comments and so would recommend minor revisions prior to publication pending the editor's assessment of whether this merits a research article or should rather be a brief communication. The contents definitely fit the scope of The Cryosphere so regardless would like to see the work published herein.

**Thank you for the positive feedback.**

Main comments:

Pending a determination of whether or not this should be considered a standalone research article or brief communication will determine whether this scope should be significantly narrowed or not

**We decided to submit a full research article in order to present both the difference between two widely used in-situ sampling techniques as well as how differences in in-situ snow depth observations might impact the specific task of validating unpiloted aerial system (UAS) based snow depth maps. It seems that we are entering an era where UAS observations may become the norm and offer the potential to improve the community's ability to capture snowpack characteristics. This improvement involves the mass of collected points, spatial coverage, and repeatability. Differences and uncertainties in in situ sampling techniques are no longer much smaller than the noise of remotely sensed estimates. We believe that a community conversation about when and how to best use these datasets to advance science is warranted. We look forward to learning the editor's thoughts.**

A bias in instrument type is evident and in this shallow snow is on the order of 12% of depth. With respect to the uncertainties evident in manual sampling and more so in the uas-lidar product is this a relevant difference in the context of all the other uncertainties involved? I'd

like to see a more direct and clear assessment of this.

**In this study, the biases are consistent, relatively predictable and thus, with sensitivity to this issue, can be accounted for. Every measurement that we have has a measure of uncertainty, but the large number of samples in this study lends credibility to our understanding of the biases. For the in-situ sampling, some errors and uncertainties are known but unavoidable such as observer bias, challenges with ice layers, and instrument performance under different conditions. López-Moreno et al.'s (2020) study demonstrates the complexity of the errors in relatively simple snowpacks with a range standard snow core samplers. An interesting finding in their study was that the interquartile range for variability in measurements, using a single instrument across local plots, is on the order of 2 to 10 cm with replicates have a coefficient of variation on the order of 0.05 (at a single location) and variation between locations resulting in a coefficient of variation on the order of 0.05 under the best conditions (snow on top of pavement) or about 2.5 cm. Our sampling strategy was designed to limit the effect of these uncertainties by sampling at many sites over a relatively small area. Likewise, at each sampling site multiple support measurements were collected to reduce the uncertainty, and thus a novelty over Berezovskaya and Kane (2007).**

**In the broader context, whether a 1-3 cm error (12%) matters depends on how the observations will be used. For most water resources applications such as reservoir operations or snow melt floods, it is unlikely to matter. For the albedo and energy balance studies where the onset of bare ground versus snow covered landscape transitions is important, these small differences likely matter because transition periods are difficult to correctly model and critical for understanding winter and spring soil processes. If the datasets are intended to serve as ground truth for modeling or remote sensing observation, these error might matter in some landsurface or watershed modeling where the goal is to appropriately characterize snow depth spatial structure and its evolution over short time scales.**

In deeper snow for example 1-3cm error is negligible…

**A goal of this research was to evaluate how sampling in a shallow snowpack can be best conducted, with the understanding that it may differ from best sampling practices in deeper snowpacks. As you mentioned, 1-3 cm error is negligible in deeper snowpacks but can be significant in shallower snowpacks that hover near the e-folding depth (ie: 3-10 cm; France et al. 2011 and others). As we mentioned in response to the other reviewer's comments, when there are shallow snowpacks it is that much more important to understand the snowpack and conditions for which an individual sampler was designed in order to select the most appropriate sampler and to develop an appropriate sampling design.**

**We plan to update section 4.1 to include a clearer discussion of the broader implications of the uncertainties as discussed in response to this comment and the previous comment.**

France, J.L., King, M.D., Lee-Taylor, J., Beine, H.J., Ianniello, A., Domine, F. and MacArthur, A., 2011. Calculations of in-snow NO2 and OH radical photochemical production and photolysis rates: A field and radiative-transfer study of the optical properties of Arctic (Ny-Ålesund, Svalbard) snow. *Journal of Geophysical Research: Earth Surface*, *116*(F4).

There is an assessment of the penetrability of the magna versus tube in leaf litter in a snow free situation. Is this relatable to the snow-covered situation. Ie compaction of leaf litter and/or snow/ice within the litter may change the penetrability of the litter versus when it is uncovered in the warm season? Any insights on this?

**The Figure R1 images from the field camera in the deciduous forest well-represent the litter prior to and after snow compaction (the beginning and end of the sampling campaign, respectively). As seen in the images, the leaf litter appears to be shallower than 10 cm in depth at both timesteps, indicating that the compaction from snow cover was minimal. Being aware of the impacts of snow cover on leaf litter compaction, we measured the litter promptly following snowmelt as to match the penetrability conditions as closely to snow-covered conditions as possible. While we did not do a full study of leaf litter penetrability in snow-covered conditions, it seems that for our forest sites that the penetrability in snow-covered conditions would be similar to the later season observations.**

[Figure]

[Figure]

**Figure R1. Field camera images in the deciduous forest on December 2, 2020 and March 17, 2021**

Is magnaprobe weight in bare scenario the same as the pressure exerted when pushed through the snow? Was there ice layering present in the snow situations that would influence the penetrability and force necessary for the probes? Were both probes pushed straight in or was the tube rotated to cut? An important distinction perhaps.

**There might be a very minor difference in the pressure exerted due to friction along the probe and snow interface. Samplers were careful to only exert the minimum amount of pressure needed for the probe to penetrate through the snowpack without exerting any excess pressure. The magnaprobe was inserted straight through the snowpack, while the snow tube was rotated to help cut through ice layers. Both instruments were held perpendicular to the snow surface.**

I don't see any clear hypothesis for why the difference between magnaprobe and snow tube? What comes to mind for me is that the tube has a wide/large surface area of orifice that will distribute pressure so wont' penetrate substrate as easily? From a scientific process perspective a hypothesis like this may streamline the content…

**While we did not state a clear hypothesis, this does reflect what we sought to test when we laid out our sampling design. We have updated the introduction to include our hypothesis in lines 109-112. Our original analysis was conducted using one-sided t-tests (see line 299).**

**We propose to add the following sentences to the last paragraph of introduction. "We hypothesize that the snow depth measurements from the magnaprobe will be deeper than those from the snow tube. This hypothesis is based on the understood errors and biases associated with each the magnaprobe and the Federal tube, including the smaller surface area of the probe which allows for greater penetration through**

**snowpacks and leaf litter."**

What about rebound of the litter/veg in bare situations? Section 4.5 in Harder et al., 2020 discusses some of the challenges associated with snow depth obs over vegetation (ie compression of vegetation by overlying snow can lead to underestimated snow depths). An additional challenge for uas-lidar is that vegetation can displace the base of a shallow snowpack from the reference soil surface introducing a bias of actual snow depth that uas-lidar and probing will both miss for example. Any of those sorts of challenges encountered? Perhaps a picture of the surfaces/leaf litter would be helpful to provide context.

**The Figure R1 images from the field camera in the deciduous forest capture the difference in the litter from snow compaction from December to March (the beginning and end of the sampling campaign, respectively). As seen in the images, the leaf litter appears to be shallower than 10 cm in depth at both timesteps, indicating that the compaction from snow cover was minimal. In Harder et al., 2020's discussion of the challenges associated with snow depth observations over vegetation, negative snow depths calculations confirmed that depths into the ground surface were being collected. Their section 4.5 indicates that these negative values were found in regions where there are shrubs and wetland reeds that bend over when loaded with snow. Our sites did not have a similar land cover that would compress and decompress.**

**Our results of shallower snowpack from airborne measurements as compared to manual measurements are largely explained by its lack of any penetration into the litter. Based upon our experience though, we broadly accept that neither the lidar nor the in-situ measurements are "true" snow depths because of leaf litter issues and that the total observed bias will include the sum of opposing biases in our measurement approaches. Over-penetrating with the magnaprobe, and (additional) compression caused by the snow tube could cause higher in-situ depths relative to truth, while a baseline elevation with loose and dry litter would have a higher elevation than the true baseline surface compressed by the weight of snow, resulting in an underestimate of snow depth relative to the truth. This bias would be anticipated to increase in land covers such as Harder et al.'s shrubs and wetland edges.**

**We intend to modify the discussion to include a more thorough discussion of the litter/vegetation challenges and to highlight this point in the abstract.**

Technical comments:

291-292- what where the absolute differences? Not as relevant a distinction if they were in very shallow snow for example.

**After checking the data for the times in which the times when the magnaprobe had measurements greater than double the depth of the snow tube measurements, we**

**found that this occurred when snowpack depth exceeded 15 cm in depth as measured by the magnaprobe (ie. not extremely shallow snowpacks). This point should be clarified in section 3.1 as follows**

**In eight pairs of measurements, when the magnaprobe measured depth greater than 15 cm, the magnaprobe snow depth was more than double the snow tube snow depth.**

381 impale - > penetrate?? The imagery of impaling leaf litter or any hapless subnival creature living there is vivid but perhaps penetrate is better/less exciting?

**The authors agree and plan to make this revision according to recommendation.**